# Research on the Environmental Effect of Green Finance Policy Based on the Analysis of Pilot Zones for Green Finance Reform and Innovations

**Haifeng Huang * and Jing Zhang**

School of Economics and Management, Beijing University of Technology, Beijing 100020, China; ZJing@emails.bjut.edu.cn
* Correspondence: crcet.huang@bjut.edu.cn

**Abstract:** In this study, taking the pilot zones for green finance reform and innovations set up in 2017 as the objects, a quasi-natural experiment is conducted to assess the environmental effects of green finance policy using the difference-in-difference propensity score matching (PSM-DID) method based on the panel data in 30 provincial-level administrative regions from 2011 to 2019. In addition, further efforts are made to investigate the differences of green financial policies in environmental effect. According to the research findings, the set-up of green finance pilot zones can reduce the environmental pollution, and green finance policy is conductive to environmental enhancement. Meanwhile, a partial mediating effect exists between a region's innovation capability and industrial structure. On the whole, green finance policy plays the most significant role in improving the eastern region's environmental pollution, followed by the central region, but barely enhances the environment in the western region. To sum up, the more serious the environmental pollution is, the better the effect of green finance policy.

**Keywords:** green finance; policy research; PSM-DID method; environmental effect; pilot zones for green finance

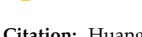



## 1. Introduction

China's environmental pressure has been multiplied by its rapid growth development model. In the Central Economic Working Conference held at the end of 2015, it was made clear for the first time that China's resource and environmental carrying capacity had reached or was close to the upper limit and that if it did not put the brakes on, it could fall into an environmental trap. The imbalance between resources environment and economic development has provoked an urgent need for green transformation of the Chinese economy. Against the backdrop of green economic restructuring, green finance is one way to propel green development. Compared with the UK, the USA, Germany, France and other developed countries in Europe and America where green finance mainly relies on the market development, China has realized its green finance development prominently characterized by top-level design. This means that the Chinese government serves as the designer and constructor of green finance system while burdening the critical mission of establishing and polishing the market mechanism and providing a favorable market environment for green finance development [1]. Considering current under-supply of green finance, lack of incentive mechanism, endogenetic deficiency and incomplete supporting facilities, the government is required to continuously strengthen the combined action of diversified policy systems involving green industrial policy, low-carbon and consumption-reduction policy and green finance policy, as a way to make existing green finance more effective [2].

Research studies on green finance policy can be carried out from two aspects: First, most research studies on the evolution of green finance policy and the construction of green

finance policy system are theoretical. For example, Aizawa et al. pointed out that, in order to tackle deep-rooted environmental problems, China launched a series of green taxation, green purchasing and finance-related green policies; among them, green credit policy was the most advanced since it displayed powerful resistance to the large-scale economic turmoil in China after the global financial crisis [3]. Drawing on the experience of the San Giorgio Group, Buchner et al. provided observations about effective green financing and specifically about how different policies and injections of public resources are already altering the behavior of private entities, financial institutions and capital markets to invest in climate change mitigation [4]. Gilchrist et al. evaluated a systematic literature survey and show that research has discovered that environmentally responsible practices not only enhance shareholder value but also the value accrued to nonfinancial stakeholders [5]. According to the research conducted by Zhang et al., green credit policy was not thoroughly brought into force as there indeed arose many problems including confusing policy details, unclear executive standards and lack of environmental information. By contrast, Jiangsu Province subtly combined green credit policy with environmental performance evaluation, which demonstrated better practicability [6]. Zhang stated that, although China took the first step in setting up the basic framework for a green finance system, specific rules were still in need of clarification [7]. From the perspective of provincial policies, Zhou et al. sorted out major policies/measures carried out by Chinese provinces to expedite green finance development in an all-round way on various aspects like green credit, green insurance, green securities, carbon finance and the founding of a green finance system. Afterwards, they revealed that, some provinces in China did not pay enough attention to green finance, and most of them did not have a sound system of green finance regulations and policies, the scale of their green finance was relatively small, the portfolio of products and services were limited, and so on [8]. According to policy changes, Chen divided the evolution of Chinese green finance policy into three stages: Early Establishment Stage, Rapid Development Stage and Differentiated Development Stage [9]. Jiang et al. affirmed the role played by the government in the development of green finance and held that the government need to act as a "visible hand" to make corresponding adjustments [10].

Second, research studies on the benefit of green finance policy are substantially empirical. Yet, research studies on the effect of green finance policy are unfolded respectively from the perspective of enterprise, commercial bank and environment. For example, in an empirical study on the data of listed companies in the manufacturing industry between 2010 and 2015, Wang et al. found that China's green finance was at an inefficient allocation level, but green policies could improve the green finance's allocation efficiency while the shortage of green supervisory policies restrained the positive impact of financial development on the green finance's allocation efficiency [11]. According to the study conducted by Yang et al., under capital rationing conditions, green credit policy could exert positive influence on renewable energy sources enterprises [12]. Liu et al. performed a quasi-natural experiment in China based on the enacting of green credit directive policy, and concluded that the debt financing capacity of heavy polluting enterprises declined sharply while state-owned enterprises and enterprises in regions with vulnerable finance ecology produced a remarkable net effect in debt financing [13]. The research findings of Niu et al. unveiled that green credit policy noticeably increased the listed company's financing convenience and reinforced the credit support for green listed companies [14]. Through an investigation into the cost efficiency achieved by 73 commercial banks in China, Ding et al. revealed that the impact of green credit policy on the cost efficiency of commercial banks took on U-type features, but it now rebounds from the U-valley bottom. Hence, in the long run, green credit policy is beneficial to upgrading the bank's cost efficiency [15]. Sun and Lei also conducted a study and found that large banks preferred to execute green finance policy which would have a positive impact on the advancement of the bank's performance and the perfecting of external financing capacity in spite of insignificant effect, further proving the imperfection of China's green finance system [16]. Based on corresponding research, Zou et al. concluded that green finance policies issued by central ministries and

commissions and local government departments were all effective in lowering the intensity of industrial pollution. Besides, the higher the synergistic degree of green finance policies between central ministries and commissions and local government departments was, the more remarkable the effect of industrial pollution reduction would be [17]. According to Batrancea's findings, in order to increase economic growth while reducing global warming and climate change, the financial sector should assume a greater role in funding green investments [18]. On the basis of carbon emissions trading pilot policies, Du et al. assessed plenty of green finance policies and summarized that pilot policies could largely reduce the carbon emission increment in pilot areas [19]. Shen et al. adopted the DID model to evaluate the effect of green finance pilot policies and spotted that green finance pilot policies could effectively bring down the energy consumption of per unit of GDP. As such policies were remarkably effective, they suggested the expansion of pilot scope [20]. Proceeding from the breakpoint regression design, Zhao et al. demonstrated the important role played by green finance policy in dramatically reducing carbon emissions of key provinces in China along the Belt and Road [21].

Although previous research studies basically mention all problems existing in the development of China's green finance policy, most scholars remain affirmative about the positive role played by it. However, for the environmental effect of green finance policy, most scholars have simply analyzed the impact or influencing mechanism of green finance policy on the environment while ignoring the specific difference in various regions. Therefore, proceeding from previous research studies, the difference-in-difference propensity score matching (PSM-DID) method is adopted to assess the environmental effect of Chinese green finance policy based on five pilot zones for green finance reform and innovations (Zhejiang, Guangdong, Guizhou, Jiangxi and Xinjiang), and then differences observed in the eastern region, central region and western region are further analyzed. At last, the quantile regression method is applied to dig out the environmental effect difference of green finance policy. Next, the literature review will be described in Part II, the introduction of research methods, variables and data in Part III, the empirical analysis and benchmark regression, mechanism and robustness analysis in Part IV, heterogeneity in Part V and conclusions and suggestions in Part VI.

## 2. Materials and Methods

### 2.1. Model Building

In this study, the difference-in-difference (DID) method is used to evaluate the environmental benefit of green finance policy. As a tool and method for assessing the treatment effect, the DID method is frequently adopted to appraise the intertemporal effect of an implemented policy. Natural experiment or quasi-natural experiment is based on the use of the DID method in which samples will be randomly or approximately divided into the Experiment Group subject to the experiment's impact and the Control Group free of policy impact. Then, through making a comparison on the difference before and after the policy implementation between Experiment Group and Control Group, the policy's effect will be assessed. In this paper, five provinces executing pilot green finance policies are classified into Experiment Group and other provinces into Control Group, in which way the quasi-natural experiment is carried out. The reference model is as follows:

$$Pollution_{it} = \alpha_0 + \beta treated_i \times time_t + \gamma Z_{it} + \lambda_i + \mu_t + \varepsilon_{it} \tag{1}$$

where $Pollution_{it}$ is an explained variable, meaning the environmental pollution index of the $i$ province in the $t$ year. $treated_i$ is a grouped dummy variable which is 1 in Experiment Group and 0 in Control Group. $time_t$ is a time-based dummy variable which is 0 before the policy implementation and 1 after the policy implementation. The cross term of $treated_i$ and $time_t$—$treated_i \times time_t$ is the explanatory variable representing the policy effect. Then, $\beta$ is the policy effect coefficient; if $\beta < 0$, it indicates that green finance policy can alleviate the environmental pollution. $Z_{it}$ is a control variable. $\lambda_i$, $\mu_t$ are respectively Fixed Region Effect and Fixed Time Effect. $\varepsilon_{it}$ is an error item.

The essence of DID estimation is to compare two groups of homogeneous samples (like ideal twin samples). When an exogenous shock appears, the DID coefficient can perfectly reflect the treatment effect researchers want to probe into as two groups of samples share the same features on other aspects. Nevertheless, it is difficult to seek completely homogeneous samples in reality and five provinces as pilot zones for green finance policy are inherently different from other provinces in many ways. Since DID variables are probably related to some potential unobservable factors, certain deviation may exist in DID coefficient estimations and it is impossible to display the true cause–effect relationship between green finance policy and environment. In this respect, if all provinces not included in pilot zones are set into Control Group in Model (1), any biased errors may arise in the model. In order to minimize such errors and guarantee a more reliable estimation, the propensity score matching method is also introduced to rematch the Control Group so as to maintain the homogeneity of samples between Experiment Group and Control Group as much as possible in every respect. Then, a DID test will be conducted again after such matching to acquire a true treatment effect.

In this study, the prevailing basic thought is to find provinces from the Control Group of non-pilot zones with the propensity score value approximately equivalent to that in the Experiment Group of pilot zones. That is to say, except that fact whether they belong to green finance pilot zones, other features of two provinces are extremely approximate. Based on this thought, we firstly need to apply Formula (2) and Logit Model to estimate each green finance pilot zone's propensity score.

$$
\begin{aligned}
Logit(treated = \;\; & 1) \\
= \;\; & \beta_0 + \beta_1 Economy_{it} + \beta_2 Population_{it} + \beta_3 Structure_{it} + \beta_4 Foreign_{it} \\
& + \beta_5 Energy_{it} + \beta_6 Innovation_{it} + \beta_7 Government_{it} + \varepsilon_{it}
\end{aligned} \tag{2}
$$

Logit Model is applied to verify whether the propensity score variable is consistent with the control variable. In order to achieve a more accurate matching, 1:1 proximity matching without replacement method is adopted. Namely, it is required to find samples with equivalent or approximate propensity score (to that in the Experiment Group) from the Control Group, and each Experiment Group can only match with one Control Group. Thus, the Control Group participating in the matching will not be subject to the secondary matching, and the Experiment Group for this year can only match with the Control Group of the year. After the matching is finished, Formula (1) will be used to re-conduct the DID estimation. In other words, the PSM-DID method will be applied to estimate the treatment effect of green finance policy on environment.

### 2.2. Data Selection

In this study, the panel data of 30 provincial-level administrative regions, including Beijing, Tianjin, Hebei, Liaoning, Shanghai, Jiangsu, Zhejiang, Fujian, Shandong, Guangdong, Guangxi, Hainan, Shanxi, Inner Mongolia, Jilin, Heilongjiang, Anhui, Jiangxi, Henan, Hubei, Hunan, Chongqing, Sichuan, Guizhou, Yunnan, Shaanxi, Gansu, Qinghai, Ningxia, Xinjiang, from 2011 to 2019 are collected and selected; Tibet, Hong Kong, Macao and Taiwan are excluded due to seriously deficient data. Data sources mainly include the China Statistical Yearbook, provincial/municipal statistical yearbook and national economy and social development statistical bulletin in various regions.

### 2.3. Variable Setting

The explained variable here is the environmental pollution index (Pollution). In this paper, the method proposed by Qu is applied to calculate the environmental pollution index according to the dynamic comprehensive evaluation principle of "horizontal and vertical" Scatter Degree Method [22]. Evaluation objects are 30 provinces/cities/regions $S_i$ ($i$ = 1, 2, ...., n) and evaluation indicators are five pollutants (sulfur dioxide, nitrogen dioxide, flue and dust, effluent volume and output of general industrial solid wastes) $X_j$ ($j$ = 1, 2, ...., m). Original data $\{x_{ij}(t_k)\}$ are acquired based on the time sequence $t_k$ ($k$ = 1, 2,

...., T). Through original data processing by using dimensionless method, the following formula can be concluded:

$$\zeta = \frac{x_{ij}(t_k) - \overline{x_j(t_k)}}{\sigma_j t_k}. \tag{3}$$

where $\zeta$ is the indicator value after the dimensionless processing; $x_{ij}(t_k)$ is the $j$ pollutant indicator at t of the $i$ province; $\overline{x_j(t_k)}$ is the average of pollutant $j$ at $t$; $\sigma_j t_k$ is the standard deviation of pollutant $j$ at $t$. The comprehensive function of time $t_k$ shall be:

$$y_i t_k \quad \sum_{j=1}^{m} \lambda_j x_{ij}(t_k), \; k = 1, 2, ...., T, \; i = 1, 2, ...., n \tag{4}$$

$y_i t_k$ is the comprehensive evaluation value of $S_i$ at $t_k$ and $\lambda_j$ is the weight coefficient. Next, the "horizontal and vertical" Scatter Degree Method will be used to calculate the weight coefficient.

By making the matrix be $H_k = X_k^T X_k$ $(k = 1, 2, ...., T)$, $H = \sum_{k=1}^{T} H_k$ shall be a $m \times m$ symmetric matrix. Through calculating the real symmetric matrix $X_k = \begin{bmatrix} x_{11}(t_k) & \ldots & x_{1m}(t_k) \\ \ldots & \ldots & \ldots \\ x_{n1}(t_k) & \ldots & x_{nm}(t_k) \end{bmatrix}, k = $

1, 2, ...., T, the weight coefficient $(\lambda_1, \lambda_2, ..., \lambda_m)^T$ can be worked out. Then, based on the weight coefficient, the comprehensive evaluation function value $y_i t_k$ can be calculated (namelym the environmental pollution index).

However, the core explaining variable is *treated* $\times$ *time*. *treated* is a grouped dummy variable which is 1 in Experiment Group and 0 in Control Group. Five provinces executing pilot green finance policies are classified into Experiment Group and other provinces into Control Group. *time* is a time-based dummy variable which is 0 before the policy implementation and 1 after the policy implementation. The pilot zones for green finance reform and innovations were set up in 2017, so *time* shall be 0 in or before 2016, but in or after 2017, it shall be 1. The cross term of *treated* and *time*—*treated* $\times$ *time* is the explanatory variable, representing the policy effect.

As a matter of fact, a series of control variables that may influence regional environmental pollution level are structured, including economic development level (*Economy*) indicated by Gross Regional Production, population size (*Population*) indicated by permanent resident population at the end of the year, industrial structure (*Structure*) indicated by the percentage of tertiary industry in Gross Regional Production, foreign investment scale *(Foreign)* indicated by the percentage of actual foreign investment in Gross Regional Production, energy consumption *(Energy)* indicated by total energy consumption, innovation capability *(Innovation)* indicated by the number of regional patent applications, and governmental intervention *(Government)* indicated by the percentage of energy-saving and environmental protection expenses in public expenditures.

## 3. Empirical Results and Analysis

### 3.1. Propensity Score Matching

In this study, a logit model is applied to conduct the nearest neighbor matching for two groups. By enabling Experiment Group and Control Group to demonstrate their differences based on the fact of whether it is in the green finance pilot zone, the Control Group after matching can simulate the environmental effect of Experiment Group not included in the green finance pilot zone. In order to ensure the result's reliability before the PSM-DID method is applied, it is necessary to conduct a hypothesis test for the balance between Experiment Group and Control Group. Namely, it is required to consider whether any significant difference can be found in concomitant variable between Experiment Group and Control Group after matching. If there is no significant difference, the PSM-DID method shall be adopted. As shown in Table 1, the balance hypothesis test result reveals that the standard error's absolute value of all matching variables after matching is less than 10%,

showing that no significant difference exists in all variables after matching. Therefore, the PSM-DID method is feasible.

In addition, the propensity score matching needs to pass the mutual support test. James et al. pointed out that mutual support could make the matching more effective and guarantee the comparability of matching samples [23]. Meanwhile, the application of mutual support test can also avoid the subset effect put forward by Lechner [24]. The kernel density functions of samples before and after matching are shown in the following Figure 1. The dotted line represents the samples of Control Group and the full line represents the samples of Experiment Group. As indicated in Figure 1, the kernel density function curve of Experiment Group can better coincide with that of Control Group after matching. With a sufficient overlapping area, the subset effect can be avoided, and the matching effect will be better.

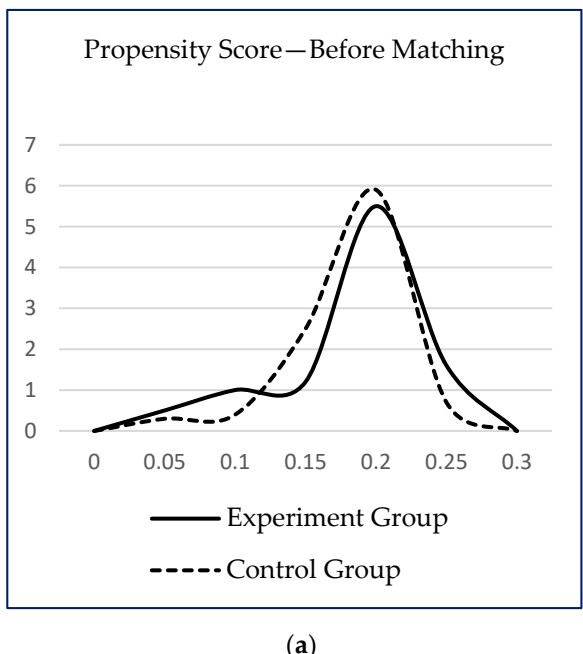

(**a**)

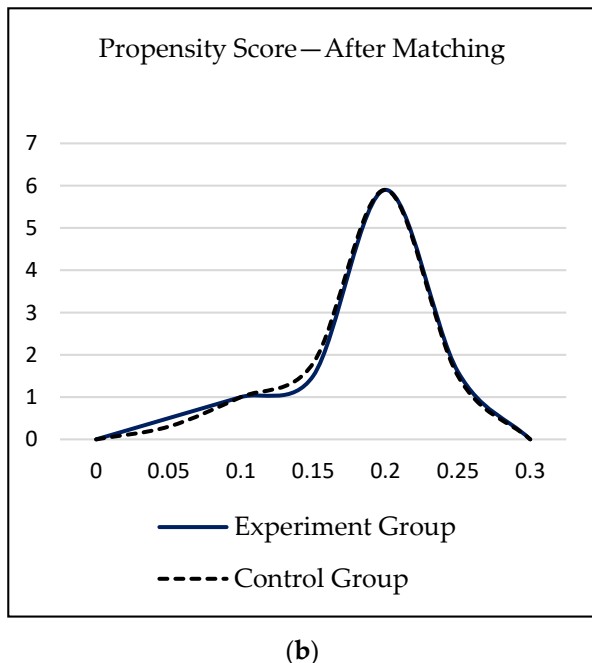

(**b**)

**Figure 1.** The kernel density functions of samples.

**Table 1.** Balance Hypothesis Test.

| Variable | Before/After Matching | Average of Experiment Group | Average of Control Group | Standard Deviation (%) | Decrease in Standard Deviation (%) | Statistical Value of $t$ | $t$-Test; $p > t$ |
|---|---|---|---|---|---|---|---|
| *Economy* | Before | 1.5536 | 1.5911 | −8.4 | 38.3 | −0.52 | 0.061 |
|  | After | 1.5536 | 1.5305 | 5.2 |  | 0.26 | 0.798 |
| *Population* | Before | 8.4571 | 8.1535 | 46.7 | 87.3 | 2.56 | 0.011 |
|  | After | 8.4571 | 8.4187 | −5.9 |  | 0.34 | 0.731 |
| *Structure* | Before | 46.659 | 48.66 | −25.7 | 95.3 | −1.37 | 0.073 |
|  | After | 46.659 | 46.566 | 1.2 |  | 0.07 | 0.945 |
| *Foreign* | Before | 0.0199 | 0.02153 | −9.6 | 65.0 | −0.62 | 0.053 |
|  | After | 0.0199 | 0.01933 | 3.4 |  | 0.17 | 0.867 |
| *Innovation* | Before | 11.038 | 10.552 | 33.3 | 87.3 | 2.14 | 0.033 |
|  | After | 11.038 | 10.976 | 4.2 |  | 0.22 | 0.826 |
| *Energy* | Before | 1.6739 | 1.4320 | 28.2 | 91.1 | 1.71 | 0.088 |
|  | After | 1.6739 | 1.6371 | 2.5 |  | 0.66 | 0.512 |
| *Government* | Before | 1.5324 | 1.3528 | 18.3 | 71.6 | 1.24 | 0.026 |
|  | After | 1.5324 | 1.4976 | 5.2 |  | 1.22 | 0.226 |

### 3.2. Analysis of Benchmark Results

After the matching of Experiment Group and Control Group is finished by using the propensity score matching method, the data after matching are subjected to the DID test; the regression results are shown in Table 2. Regression results without considering other factors are displayed in Column 1 and regression results containing control variables are displayed in Column 2. According to corresponding regression results, the effect coefficient of green finance policy is negative when no control variable is added, and it proves significant at the level of 1%, meaning that green finance policy indeed reduces environmental pollution. After various variables are controlled, including economy, population and industry, the policy effect coefficient is still negative, and it proves significant at the level of 1% with greater absolute value of the coefficient, indicating that green finance policy can better alleviate environmental pollution.

**Table 2.** Benchmark Regression Results.

| Variable. | (1) | (2) |
|---|---|---|
| *Treated × time* | −0.2290 *** | −0.5219 *** |
|  | (0.1237) | (0.3088) |
| *Economy* |  | 0.2488 ** |
|  |  | (0.3125) |
| *Population* |  | 0.9411 *** |
|  |  | (0.2554) |
| *Structure* |  | −0.0567 *** |
|  |  | (0.0115) |
| *Foreign* |  | −3.3702 * |
|  |  | (3.3244) |
| *Innovation* |  | −0.6157 *** |
|  |  | (0.1360) |
| *Energy* |  | 0.0015 *** |
|  |  | (0.0008) |
| *Government* |  | −0.0010 *** |
|  |  | (0.0009) |
| _CONS | 3.2758 *** | 0.9158 *** |
|  | (0.0652) | (1.2392) |
| Regional Effect | YES | YES |
| Time Effect | YES | YES |
| N | 270 | 270 |
| $R^2$ | 0.0146 | 0.0034 |

Note: (1) *, **, *** mark significance at the level of 10%, 5% and 1%, respectively; (2) in parentheses is the value of the standard error.

Among various control variables, economic development level, population size and energy consumption will remarkably aggravate the environmental pollution, while industrial structure, foreign investment scale, innovation capacity and governmental intervention can largely relieve environmental pollution. According to the Environment Kuznets Curve, the advancement of economic development level will trigger off the pollution risk of industrialization (including the discharge of sulfur dioxide, nitrix oxide and particulate matters) to undergo a rise first and then drop. Accordingly, the economic growth will pose a higher risk of environmental pollution, reflecting that China's economy is on the rise and the economic development level mainly depends on the industrial development level. At the same time, a majority of China's population are concentrated in provinces realizing a higher degree of economic development, which has resulted in heavier industrial pollution and domestic pollution. It turns out that the increase in population size will worsen the environmental pollution. The higher the energy consumption, the heavier the environmental pollution. As the tertiary industry mainly consists of finance, transportation and other low-pollution and low-consumption sectors, the development of tertiary industry symbolizes lower level of environmental pollution. Moreover, innovation capacity represents a region's hi-tech state. Technical innovations initiated by various enterprises can improve the comprehensive utilization of "three industrial wastes", add more value to products and effectively raise the industrial $SO_2$ removal rate while reducing the industrial carbon emission intensity [25,26]. Hence, scientific and technological innovations are conducive to the improvement of environmental pollution. Although this paper upholds the opinion that the foreign investment scale can significantly reduce the environmental pollution, previous research studies remain controversial about the impact of foreign investment on the invested country's environment. In particular, there are three mainstream hypotheses: The first is the Pollution Haven Hypothesis. The hypothesis supports that foreign direct investment ( FDI) inflow will exacerbate the invested country's environmental pollution. In the early days of industrialization, developing countries preferred to lower their environmental standards to attract foreign investment for the purpose of quickening the development of local economy, which catered for developed countries' needs in reducing the cost of pollution control. For this reason, developed countries gave priority to transferring their polluting industries and industrial chain to other regions with looser environmental regulations, and such regions were liable to be Pollution Havens. The second is the Pollution Halo Hypothesis. The opinion holds that FDI will exert positive impact on the local environmental quality. That is to say, the advanced cleaner production technology brought by FDI will not only lessen the pollutant discharge, but also stimulate local enterprises' green production by means of demonstration, competition and learning effect. Furthermore, it can boost the whole industry's pollution control efficiency and enhance local environmental welfare. The third is that the relation between FDI and the invested country's environmental pollution is non-linear [27]. Therefore, the impact of foreign investment on the invested country's environment is relatively complicated, constituting a key variable to be discussed in this paper. Unnecessary details will not be unfolded here.

### 3.3. Analysis of Influencing Mechanism

The experimental result shows that the implementation of green finance policy will ameliorate the environmental pollution. Then, what is the mechanism of improving environmental pollution based on green finance policy? In reality, the implementation of green finance policy will impose restrictions on enterprises' frequent pollution behaviors and facilitate more capital to flow into low-pollution industries, in which way the industrial structure can be optimized to realize the environmental enhancement. It is the role of green finance policy in restricting enterprises' behaviors and the increase in green funds that have stimulated enterprises' technological innovation capacity. Hence, in this paper, innovation capacity or industrial structure is treated as mediating variable which will be involved in the mechanism of green finance policy's impact on environment. The stepwise regression method is adopted here. First of all, the environmental pollution

index is subject to the regression based on *treated* × *time*. However, the difference lies in that innovation capacity or industrial structure is a mediating variable rather than a control variable. If the policy effect coefficient is significantly negative, green finance policy becomes effective in reducing the environmental pollution. Then, innovation capacity or industrial structure will be subject to regression based on *treated* × *time*. If the policy effect coefficient is significantly positive, green finance policy becomes effective in enhancing innovation capacity or perfecting industrial structure. Lastly, the environmental pollution index shall be subject to the regression based on *treated* × *time* and industrial structure or innovation capacity. If innovation capacity or industrial structure coefficient is significantly negative and the policy effect coefficient is not significant or is significant with a decrease in the absolute value, it proves the existence of a mediating effect. The model can be set as follows:

$$Pollution_{it} = \alpha_0 + c \times treated_i \times time_t + \gamma Z_{it} + \varepsilon_{it} \tag{5}$$

$$M_{it} = \alpha_0 + a \times treated_i \times time_t + \gamma Z_{it} + \varepsilon_{it} \tag{6}$$

$$Pollution_{it} = \alpha_0 + c\prime \times treated_i \times time_t + b \times M_{it} + \gamma Z_{it} + \varepsilon_{it} \tag{7}$$

As shown in Table 3, the first three columns reflect the regression results by regarding innovation capacity as the mediating variable, and the last three columns the regression results by regarding industrial structure as the mediating variable. According to corresponding results, if innovation capacity acts as a mediating variable, the $c$ value is significantly negative, the value of $a$ is positive, those of $c\prime$ and $b$ are negative, and the absolute value of $c\prime$ is smaller than that of $c$, meaning that partial mediating effect exists in an innovation capacity; if industrial capacity is treated as a mediating variable, the $a$ value is significantly positive, the $c\prime$, $c$ and $b$ values are negative, and the absolute value of $c\prime$ is smaller than that of $c$, proving that a partial mediating effect also exists in industrial structure.

**Table 3.** Mechanism Analysis Results.

| Variable. | (3) | (4) | (5) | (6) | (7) | (8) |
|---|---|---|---|---|---|---|
| *Treated* × *time* | −0.6643 *** | 0.1070 * | −0.5219 *** | −0.5438 ** | 0.0250 * | −0.5219 *** |
| | (0.2885) | (0.0676) | (0.3088) | (0.3049) | (0.0249) | (0.3088) |
| *Structure* | | | | | | −0.0567 *** |
| | | | | | | (0.0115) |
| *Innovation* | | | −0.6157 *** | | | |
| | | | (0.1360) | | | |
| Control Variable | YES | YES | YES | YES | YES | YES |
| Regional Effect | YES | YES | YES | YES | YES | YES |
| Time Effect | YES | YES | YES | YES | YES | YES |
| N | 270 | 270 | 270 | 270 | 270 | 270 |
| R² | 0.8094 | 0.8894 | 0.0034 | 0.8250 | 0.5155 | 0.0034 |

Note: (1) *, **, *** mark significance at the level of 10%, 5% and 1%, respectively; (2) in parentheses is the value of the standard error.

### 3.4. Robustness Test

In addition to the establishment of pilot zones for green finance policy, other policies or random factors will also affect the environmental conditions. In order to avoid the influence exerted by these factors, the counter-factual test is adopted in this paper similarly to previous research studies [28]. It is assumed that green finance pilot zones were set up in 2015 before the regression was performed. If the then-policy effect coefficient was significantly negative, it revealed that the relief in environmental pollution did not benefit from the setup of green finance pilot zones, but changes in other policies or random factors triggered such reduction; if the policy effect coefficient was not significantly negative, it indicated that the establishment of green finance pilot zones mitigated the environmental pollution. As shown in Table 4, when the year of establishment was changed to 2015 or after 2015, the policy effect coefficient was negative, but it was not significant with the coefficient's absolute value being smaller than the benchmark regression result, which proved

that the improvement of environmental pollution was mainly driven by the establishment of green finance pilot zones and this verified the result's reliability.

**Table 4.** Robustness Test Results.

| Variable | Counter-Factual Test |
|---|---|
| *Treated × time* | −0.1412 |
| | (0.3296) |
| _cons | 3.2681 |
| | (0.2032) |
| Control Variable | YES |
| Individual Effect | YES |
| Time Effect | YES |
| N | 270 |
| $R^2$ | 0.0002 |

Note: in parentheses is the value of the standard error.

## 4. Difference Analysis

### 4.1. Regional Heterogeneity Analysis

The vast territory of China generates greater differences in the development of the eastern region, central region and western region. Accordingly, the influence of green finance policies on environmental pollution varies across regions, as well. For this reason, it is necessary to analyze the regional heterogeneity so as to obtain more accurate results. Meanwhile, five green finance pilot zones are equally distributed in eastern region, central region and western region of China, which is beneficial to the regional analysis. Consequently, the eastern region, central region and western region are subject to the regression, and corresponding results are shown in Table 5. On the whole, each region's result is roughly consistent with the nationwide result. A negative policy coefficient further demonstrates the positive impact of green finance policy on environmental enhancement. The eastern region and central region are separately significant at the level of 1% and 5% while the score for the western region is not significant. Compared with the central region and eastern region, the western region lags behind in terms of financial development and is not inherently advantageous on various aspects such as talents, capital and infrastructures. As a result, financial functions cannot bring out unusual results; the demand for green finance is smaller, and it is extremely difficult to realize the project financing. To sum up, a huge funding gap exists in the green finance development of the western region, which prevents green finance policy from achieving the ideal effect. Therefore, the policy effect achieved in the western region is not significant.

**Table 5.** Regional Heterogeneity Analysis Results.

| Variable. | Eastern Region | Central Region | Western Region |
|---|---|---|---|
| *Treated × time* | −0.45196 *** | −0.4181 ** | −0.3149 |
| | (0.2281) | (0.2301) | (0.2769) |
| _cons | 4.5845 *** | 1.9253 *** | −2.0579 *** |
| | (3.0940) | (0.0022) | (1.0241) |
| Control Variable | YES | YES | YES |
| N | 108 | 81 | 81 |
| $R^2$ | 0.9392 | 0.8602 | 0.8466 |

Note: (1) **, *** mark significance at the level of 5% and 1%, respectively; (2) in parentheses is the value of the standard error.

### 4.2. Quantile Regression Analysis

In fact, the benchmark regression result has attested the positive effect of green finance policy in easing environmental pollution, and such an effect is even more significant in the central region and eastern region. In order to perform a more profound analysis, the quantile regression method is applied to verify the difference in the environmental effect

of green finance policy. Corresponding regression results are shown in Table 6. In the 0.75–0.9 quantile, the policy effect coefficient is significantly negative, meaning that green finance policy indeed eases environmental pollution; in the 0.1–0.5 quantile, the policy effect coefficient is negative, but not significant; the absolute value of policy coefficient increases as the quantile increases, reflecting that the policy effect of green finance is more significant in regions producing more pollution. For low-pollution regions, the effect of green finance policy is insignificant, which explains the insignificant policy coefficient of green finance policy in the western region. Although environmental pollution in the central region is greater than that in the eastern region, the effect achieved in the eastern region is superior to that in the central region. Possible reasons can be found from two aspects: I. In the eastern region, there exists a huge pollution gap. Although the central region as a whole is at a higher level of pollution than the eastern region, several areas reaching the highest level of pollution are distributed in the eastern region. Therefore, the policy effect achieved in the eastern region is superior to that in the central region; II. The eastern region reaches a higher financial level than the central region. Especially in Beijing, Shanghai and other cities featuring solid foundation and a higher level of development, it is more suitable for green finance policy to yield unusually brilliant results.

**Table 6.** Quantile Regression Results.

| Variable. | QR10 | QR25 | QR50 | QR75 | QR90 |
|---|---|---|---|---|---|
| *Treated* × *time* | −0.1449 | −0.1612 | −0.1816 | −0.2171 ** | −0.2355 *** |
| | (1.9138) | (1.5049) | (1.0187) | (0.5541) | (0.7905) |
| Control Variable | YES | YES | YES | YES | YES |
| N | 270 | 270 | 270 | 270 | 270 |

Note: (1) **, *** mark significance at the level of 5% and 1%, respectively; (2) in parentheses is the value of standard error.

## 5. Conclusions and Prospects

China's green finance system possesses especially noticeable top-level design characteristics, and some achievements were attained by virtue of the top-down development mode in the last few years. In June 2017, the State Council decided to set up pilot zones for green finance reform and innovations with their own highlights and features respectively in Zhejiang, Guangdong, Guizhou, Jiangxi and Xinjiang, marking a new chapter for China's green finance entering the innovative practice stage. In this paper, a quasi-natural experiment is carried out based on pilot zones. According to the panel data of 30 province-level administrative regions from 2011 to 2019, the PSM-DID is applied to evaluate the environmental effect of green finance policy. Corresponding research findings indicate that: (1) the establishment of green finance pilot zones has reduced the environmental pollution and green finance policy is conducive to environmental enhancement. Moreover, a partial mediating effect exists between a region's innovation capacity and industrial structure [29]. (2) In summary, the effect of green finance policy in easing the environmental pollution of the eastern region is the most obvious, followed by the central region. Yet, the effect achieved in the western region is insignificant. (3) The heavier the environmental pollution, the better the green finance policy's effect. On the basis of these conclusions, the following suggestions are proposed:

Firstly, the scope of pilot zones for green finance reform and innovations shall be expanded. Proceeding from regions at the highest level of pollution, such expansion shall gradually cover the whole country. Meanwhile, the regional difference shall also be taken into consideration and adjustment shall be made in line with local conditions. The local government shall encourage local financial institutions to cooperate with large-scale financial institutions and carry out special training so as to materialize the innovative design, development and utilization of green finance products in line with the local situation.

Secondly, it is required to polish the green finance policy system, set up the specific incentive mechanism, intensify the support from fiscal policies and preferential tax policies

(including the government's interest subsidies, guarantees and premium subsidies), and improve the basis for green finance-related laws and regulations. The enforcement of environmental-protection laws should be enhanced, relevant entities' rights and obligations should be defined, and high-polluting enterprises should be required to purchase environmental pollution liability insurance. Efforts should also be made to reinforce the information sharing and disclosure mechanism, include enterprises' pollution discharge and energy consumption information into the national credit information sharing and exchange platform, and add the article requiring listed companies and bond-issuing enterprises to disclose environmental information into Securities Law. Furthermore, the local government shall take positive action and work out local Green Industry Guidance Catalogue based on regional advantages and actual industrial situation [30].

Thirdly, the green rating system should be established. Usually, GDP is frequently applied to measure the economic development level, which will result in the excessive pursuit of beautiful numbers, while ignoring various environmental problems. To some extent, Green GDP can be applied to avoid such problems. When we are introducing foreign investment, corresponding green evaluation standards shall also be formulated to raise the foreign investment access threshold. In order to upgrade the green rating and green index, investors shall attach more importance to the green evaluation on enterprises [31].

Fourthly, effort shall be made to heighten the local green finance level, strengthen the construction of green finance infrastructures, create the agglomeration effect by introducing various financial institutions and evoke various financial institutions to establish their green finance BU to promote the development of local green finance. Besides, it is necessary to cultivate skilled works and urge practitioners to attend green finance-related training classes, attract talents by providing them with preferential policies and actively cooperate with colleges and universities and research institutes to retain talents. Additionally, great importance shall be attached to improving the investors' awareness of green investment, encourage large-scale financial institutions to release more green finance products, and arouse greater market demands, which is directly to the benefit of the sustainable development of financial institutions.

There are limitations to this study that must also be recognized. Firstly, we adopt DID in our study, but the conditions for DID are relatively harsh. Although we introduce t the propensity score matching method and the kernel density function curve of Experiment Group can better coincide with that of Control group after matching, the number of control variables is limited after all, so we cannot guarantee that Experiment Group and Control Group are completely homogeneous except for the fact of whether they belong to green finance pilot zones. In addition, our study is based on the macro level, without considering enterprises and financial institutions, and without subdividing the green financial policy, such as green credit policy and green securities policy, which are also the direction of our study in future.

**Author Contributions:** Conceptualization, H.H. and J.Z.; methodology, H.H.; software, J.Z.; data curation, J.Z.; writing—original draft preparation, J.Z.; writing—review and editing, H.H. All the authors contributed to drafting the manuscript and approved the final version of the manuscript. All authors have read and agreed to the published version of the manuscript.

**Funding:** This research received no external funding.

**Institutional Review Board Statement:** Not applicable.

**Informed Consent Statement:** Not applicable.

**Data Availability Statement:** Publicly available datasets were analyzed in this study. This data can be found here: National Bureau of Statistics. Available online: http://www.stats.gov.cn/ (accessed on 2 August 2020).

**Conflicts of Interest:** The authors declare no conflict of interest.

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
