# Peer review of "Research on the Environmental Effect of Green Finance Policy Based on the Analysis of Pilot Zones for Green Finance Reform and Innovations"

_sustainability, doi:10.3390/su13073754_

Round 1

Reviewer 1 Report

Dear Authors,

I believe you have touched a very interesting topic by focusing on the effect of green finance policy. Topics within this framework certainly need to be investigated and examined further to expand scientific as well as practical knowledge. The manuscript needs various revisions in terms of content and formatting, in accordance with the academic standards of the journal. Please find these comments in the following paragraphs.

The article needs thorough English proofreading to improve various shortcomings. For instance, authors should not use contractions such as “don’t” or “isn’t” but use “do not” and “is not”. Moreover, authors should pay attention to the use of the comma (e.g., “in which way, the “quasi-natural experiment”; “need pass the mutual support test” should be worded as “needs to pass…”), they should make the agreement between the subject and the verb. There are additional aspects regarding missing prepositions, incorrect use of nouns instead of corresponding adjectives that need to be amended.   

My main shortcoming of the paper is that the mathematical models lack clear explanations and consistency, including the variables included in the models. For instance, on line 136, the mathematical formula of the model comprises “time” as a variable, but when explaining the variables the authors identify the variable “time” as “policy” (as it appears). If this is not the case, and the variables “time” and “policy” should be both included in the model, this aspect has to be remedied. Moreover, authors mention that “B is the policy effect coefficient”, but policy is not included in the model. I suggest authors to carefully check the accuracy of all mathematical models, give clear and straightforward explanations on which variables are included in the model and why.  

Abstract: “partial mediating effect is existent between…”. This should be rephrased as “partial mediating effect exists…”.

Line 27: “If we don’t push STOP button”. The wording is not adequate, please remove “STOP button” and rephrase.

Line 126: What do you mean by “utility method”?

Line 211: “…variable is treated x time…”. What is the meaning of the variable “treated x time” and where is this included? It is nowhere to be found.    

Authors use extensively the word “researches” but this should be replaced with “research studies” for more adequate wording.

Please use an adequate in-text citation style. Even if you did specify the number of the reference, you included authors’ first and last name. Please use only authors’ last names.

Line 163: “firstly apply”. Please add “to” after the word “firstly”.  

Lines 334-336: Are these three different models? In the sentence prior to them, you mentioned the following: “The model can be set as follows:”. Please clarify this aspect.  

Please expand the reference list because it is rather scarce. Therefore, I kindly ask you to include at least the following sources, among others:

  • Gilchrist, D., Yu, J., & Zhong, R. (2021). The Limits of Green Finance: A Survey of Literature in the Context of Green Bonds and Green Loans. Sustainability, 13(2), 478.
  • Batrancea, I., Batrancea, L., Rathnaswamy, M.M., Tulai, H., Fatacean, G., & Rus, M.-I. (2020). Greening The Financial System in USA, Canada and Brazil: A Panel Data Analysis. Mathematics, 8(12), 2217.
  • Buchner, B.K., Heller, T.C., & Wilkinson, J. (2012). Effective Green Financing: What Have We Learned So Far? Climate Policy Initiative.

Which are the limitations of the study? I kindly ask you to mention them in the concluding section.

Please include a list with all 30 provinces/cities/regions on which you ran the analyses. This aspect is missing from the current version. 

Author Response

Dear reviewer:

Thank you very much for your evaluation and comments on our paper . We have revised the manuscript according to your kind advices.Please see the attachment. Thank you very much for all your help and looking forward to hearing from you soon.

Best regards, 

Haifeng Huang

Reviewer 2 Report

Thank you for giving me the opportunity to read and reflect on your paper, I think this is well structured and well written. It is a good paper.

However in my opinion, some important details should be emphasized in order to improve the quality of the research before publishing:

  • You describe the panel data of 30 domestic provinces/cities/regions. How were they chosen? Are they randomizend chosen? And why 30? I am concerned that the sample size is not sufficiently representative of population size for a study of these characteristics. Please add some detail.
  • The Conclusions and Suggestions section, should connect the research results with relevant literature citations for validity and reliability.

This are the onlys points i would like to clarified in the paper

Author Response

Dear reviewer:

Thank you very much for your evaluation and comments on our paper . We have revised the manuscript according to your kind advices. Enclosed please find the responses to you.  Thank you very much for all your help and looking forward to hearing from you soon.

Best regards, 

Haifeng Huang

Round 2

Reviewer 1 Report

Dear authors,

The revised version of the manuscript is considerably improved as compared to the previous version. I only have only observation. In the section containing limitations, you used a contracted term "can't". I suggest you replace it with "cannot", since it is more adequate in a scientific writing. 

Author Response

Dear reviewer:

Thank you for your comments and suggestions, which are valuable in improving the quality of our manuscript. We have revised the manuscript according to your kind advices. Please see the attachment. Thank you very much for all your help and looking forward to hearing from you soon.

Best regards,

Haifeng Huang
